# A Qualitative Study on the Position and Role of Volunteers in Integrated Care—An Example of Palliative Care in Croatia

**DOI:** 10.3390/ijerph19138203

**Published:** 2022-07-05

**Authors:** Dorja Vočanec, Karmen Lončarek, Maja Banadinović, Slavica Sović, Aleksandar Džakula

**Affiliations:** 1Department of Social Medicine and Organization of Health Care, Andrija Štampar School of Public Health, University of Zagreb School of Medicine, Rockefeller St. No. 4, 10000 Zagreb, Croatia; adzakula@snz.hr; 2Department of Integrated and Palliative Care, Rijeka University Hospital Center, Krešimirova St. No. 42, 51000 Rijeka, Croatia; karmen.loncarek@uniri.hr; 3Firefly Association, Prilaz Gjure Deželića No. 50, 10000 Zagreb, Croatia; maja@krijesnica.hr; 4Department of Medical Statistics, Epidemiology and Medical Informatics, Andrija Štampar School of Public Health, University of Zagreb School of Medicine, Rockefeller St. No. 4, 10000 Zagreb, Croatia; ssovic@snz.hr

**Keywords:** volunteers, integrated care, palliative care

## Abstract

Volunteers have been present in palliative care since its inception. With the development of palliative care systems, their role and position are changing. Given growing long-term care needs and limited resources in health and social care, volunteers are becoming an important resource in meeting these needs. In Croatia, palliative care has been developing as an integrated care model since 2014. To assess the position and the role of volunteers, we analyzed legislative documents from healthcare and social care and conducted a focus group with volunteers in palliative care. We found that volunteers provide support from the social aspect of care, for the patient and the family. The formal palliative care system involves them as partners in the provision of care, even though this cooperation is informal. The main determinants of their activities are an individualized approach, flexibility, a community presence, and project funding. In conclusion, these determinants allow them to react quickly to identified needs, but with them come some uncertainties of their sustainability. Their activities could indicate what needs to be integrated between health and social care and in what areas. Volunteers both fill in gaps in the system and are ahead of the system, and by doing this they develop new processes around identified unmet needs.

## 1. Introduction

Caring for the dying has long been widely embedded in the family and community. Palliative care also began to develop based on the need to help in situations where formal and professional care could not provide comprehensive care, that is, when the dying were left to fend for themselves. In addition to launching the hospice movement in England, Cicely Saunders also laid the foundations for a comprehensive approach in which informal forms of care, including volunteers, are integrated into ecosystem of care [1,2,3,4].

Positive experiences of the hospice movement, as well as the development of medicine and healthcare, brought back care to the healthcare and social care systems. This time increasingly respecting the importance of all forms of care and all stakeholders. Since the beginning of this millennium, efforts have been made to integrate palliative care in healthcare systems [5,6]. WHO defines palliative care as “an approach that improves the quality of life of patients (adults and children) and their families who are facing the problems associated with life-threatening illness, through the prevention and relief of suffering by means of early identification and correct assessment and treatment of pain and other problems, whether physical, psychosocial or spiritual” [7]. Thus, palliative care is deemed an example of comprehensive care that considers different aspects of needs. These principles of integration are further seen in long-term care that does not separate a person’s medical and social needs, and requires their integration in the development of services in this area [8]. Despite the progress that has taken place in many countries, there is still a global mismatch between palliative care needs and the availability of resources and services to meet those needs. In countries where the system has not (fully) responded to the needs of palliative patients, by providing financial, human, material or other resources, palliative care still relies on the help of charities [6].

In Croatia, volunteer activities in the field of palliative care began during the 1990s. These activities were focused on patients visited at their home, but also on educating the public and professionals about what palliative care is [9,10]. The legal framework for the organization of palliative care was created in 2003. However, significant development occurred only with the adoption of the strategy on palliative care in 2014 [11]. This represented a system-level intervention, based on existing elements of healthcare, the results of which are visible in the field of resource development, the regulation of activities, and the professionalization of palliative care [12]. 

Volunteers are people who by free will commit their time and energy for the benefit of another person or community. At the same time, they do not receive financial compensation for it [13,14]. What distinguishes volunteers from informal carers is the lack of a social relationship between the volunteer and the person they provide care for [15]. This paper further deals with formal volunteers, i.e., those gathered by non-governmental non-profit organizations (NGOs) [16]. NGOs employ professionals whose job is to recruit, educate, coordinate, and supervise volunteers, to maintain continuity and quality [17]. Volunteers provide care in different settings, including hospices, hospitals, care homes, and patients’ homes. The activities they carry out are often divided into those aimed directly at the patient, and all others, including assistance in administration, fundraising, cooking, family support, and more [15,18,19,20]. The extent of the involvement of volunteers in palliative care varies from state to state, and even within individual states, where volunteers are generally more accessible in larger urban areas. Their role depends on the organization of palliative care, and health and social regulations. With the development of systematic palliative care, the role of volunteers is also changing [21].

Given the demographic trends, growing long-term care needs, and limited resources in health and social care, volunteers have become the third pillar that some governments count on in meeting long-term care needs, and therefore reducing inequalities and achieving sustainability of care [15,22,23]. In the light of that, the European Association for Palliative Care emphasized the need for the recognition, promotion, and support of volunteer work by developing a charter on volunteering in hospice and palliative care, in which Croatia also participated [24]. However, there are some limitations to its practical use [25].

The aim of this paper is to present the position and role of volunteers in the processes of palliative care in the Republic of Croatia, and to assess their capabilities and limitations in the development of the palliative care system.

## 2. Methods

In the first phase of the research, we analyzed legislative documents (laws, regulations, strategies) regulating the activities of volunteers and NGOs that gather them. We observed the position of volunteers in healthcare and in social care system, specifically in palliative care, i.e., care for the seriously ill. This included how their involvement in the care process is envisaged, what criteria they need to meet, and which activities they can carry out.

The following documents were included in the analysis: Act on Volunteering, Health Care Act, Plan and Program of Health Care Measures 2020–22, Strategic Plan for the Development of Palliative Care 2014–2016, National Program for the Development of Palliative Care 2017–2020, National Development Strategy until 2030, National Health Development Plan for the period from 2021 to 2027, Social Care Act, Ordinance on minimum conditions for the provision of social care [26,27,28,29,30,31,32,33,34]. 

In the second phase of the research, a focus group was conducted with volunteers in palliative care. According to data from the National Program for the Development of Palliative Care 2017–2020, 16 volunteer organizations operate in the field of palliative care [34]. A valid contact (phone number and e-mail address) of a total of 14 volunteer organizations was found by internet search. They were sent an e-mail invitation to participate in the focus group, and were then contacted by phone. Finally, seven volunteer organizations responded to the invitation in the affirmative. Their representatives (5 women and 2 men) were participants in the focus group. Regarding the region of their activity, three NGOs were from the city of Zagreb, two from eastern Croatia, and one each from northern and southern Croatia. Criteria for selecting participants within the volunteer organizations included that they had volunteer experience in palliative care of more than a year. In addition, each of the participants was actively involved in the management of the volunteer organization (in the position of president, vice president of the NGO, volunteer coordinator, etc.).

The research was conducted within the “Health Observatory” project (UP.04.2.1.06.0045), and as part of the doctoral dissertation “Determinants of the long-term care integration process in the Republic of Croatia based on a palliative care model”. It was approved by the Ethics Committee of the School of Medicine, University of Zagreb (REG. NO: 380-130/134-21-2). Participants were informed about the goals, benefits, and risks of the research, and had voluntarily agreed to participate by signing the informed consent. The identity of focus group participants is protected and known to researchers. Participants were introduced to the way the focus group was to be conducted, as well as the fact that they could opt out of the research at any time by informing the focus group moderator.

The focus group was conducted via Zoom, lasting 90 min. The moderator asked pre-structured questions and facilitated the discussion. The questions were developed and agreed among the authors, based on the understanding of the context in line with the findings of document analysis and previous research on palliative care in Croatia (11, 12). Along with the moderator, authors 1 and 3 observed the conversation and took notes. A transcript of the conversation was made from the audio-visual recording, which was subjected to a qualitative analysis.

The explanatory analytical approach was used, guided by research domains: (a) volunteer activities in palliative care; (b) the process of involving volunteers in palliative care; (c) use of resources in palliative care; (d) the relationship between informal and formal care provision. It is a combination of the inductive and deductive approach and is focused on the interpretation of and giving meaning to the research [35]. For the purpose of data analysis, we used thematic analysis as described by Braun and Clarke [36]. First, the authors read the transcript carefully and became acquainted with the data. Then we highlighted sections of text and labeled them based on their content, thus generating codes. Next we collated all the data within each code to obtain a clearer overview of the data and discussed obtained codes, meaningful for our research question. The codes were then grouped until agreement of the themes was reached. In the presentation of the findings, we analyze and interpret the generated codes and themes matched with selected quotes that best illustrate them, each labelled with a participant number.

For reporting, we used Appendix A [37].

## 3. Results

By reviewing the documents listed in the research methodology, the following determinants of palliative care volunteers were identified:

In Croatia a special law defines a volunteer as a person who voluntarily invests personal time, effort, knowledge, and skills to perform services or activities for the benefit of another person or community and does not receive financial or any other property compensation for it. Volunteering may be organized by any not-for-profit legal entity and must be focused on common goods. A ministry governing volunteers is the Ministry of Labor, Pension System, Family and Social Policy, also responsible for organizing the social care system.

In the Social Care Act, volunteering is mentioned in the context of the obligation of social care institutions to encourage and develop it. In addition, it recognizes NGOs (which could also apply to volunteers as NGOs organize them) as partners in the preparation, adoption, and implementation of social programs, within the combined social policy.

The ordinance on the minimum conditions for the provision of social care does not recognize palliative patients specifically; however, there is a category of elderly and severely ill persons, who are entitled to housing services as permanent stay (e.g., in nursing home), day care, or at home services. However, conditions for recognizing the right to at home services, including organizing meals, completing household chores, maintaining personal hygiene, and performing other daily needs, are based on the money income census, which excludes many palliative patients with such a need. Psychosocial counselling services for individuals and/or families are aimed at overcoming difficulties related to old age and other adverse conditions, based on an assessment of the risks, strengths, and needs in accordance with the individual change plan. Here seriously ill persons are not explicitly recognized as beneficiaries, so it would be necessary to check whether there are possibilities for exercising that right in practice.

The Health Care Act specifically recognizes volunteers as stakeholders in the organization and implementation of palliative care. Accordingly, in the Plan and Program of Health Care Measures, in addition to the providers defined as team leaders, volunteers are defined as associates who participate in the direct implementation of the procedures at the primary level of healthcare. In the strategic documents related to the development of palliative care, they are recognized as stakeholders at the level of specialist support for general palliative care, providing care in acute hospitals, long-term care hospitals, and home care. Their education is planned, as well as their inclusion in the provision of palliative care, including through the information system. It is expected that volunteers contribute to the quality of palliative care, especially to its development in the local community, and to help raise public awareness of palliative care. The coordination and integration of all care providers, especially between the healthcare and social care system is among the goals of the National Development Strategy and the National Health Development Plan.

Thematic analysis of the focus group transcript identified the following themes determined by grouping the codes identified in the participants’ statements.

### 3.1. The Role of Volunteers in Palliative Care

The activities carried out by volunteers in palliative care were determined from the testimonies of focus group participants. They are grouped into the seven codes interpreted below (Table 1), with illustrative quotations. In general, the activities they carry out are aimed at the patient and their family, the public, and at the members of the NGO themselves, i.e., volunteers in palliative care. Activities aimed at the patient and their family are from the field of social care, while activities aimed at citizens have elements of public health interventions.

#### 3.1.1. Psychosocial Support

Psychosocial support in this context refers to empowering a person to successfully cope with stress, fear, sadness, etc., and adjusting functioning in a new situation caused by a serious illness or death. Forms of providing psychosocial support by volunteers are diverse: from socializing with the patient in his home, through individual and group counselling, to art workshops and recreational meetings. Volunteers make the conversation that doctors fail to have with patients facing incurable disease and their families due to the time constraints.


*“We are mainly based on psychosocial assistance to patients and their family members. Therefore, I am actively providing psychological counselling, which is quite branched and has many aspects. And the way we provide it goes from individual counselling, group counselling, to art therapy, occupation workshops, home visits, recreational meetings and so on.” (Participant 1)*


#### 3.1.2. Assistance in Exercising Rights

Volunteers describe the confusion of patients and families at the time they are informed of the incurability of the disease or discontinuation of curative treatment. It is manifested in ignorance of the existing resources and rights they have, and in ignorance of the impending processes. Volunteers guide them in this regard by providing information and advice, and for some populations they directly advocate for the exercise of rights at the competent institutions.


*“Our two non-medical volunteers advise, that is, we refer patients to them when they are discharged from the hospital, what they can do, what their patients’ rights are and so on… Usually people call us a lot.” (Participant 5)*



*“When the child dies, somewhere at the hospital, there is no employee sitting there to tell the parent: “Parents, you must do this and that now.” Here, parents simply get the documentation and are told to organize a funeral. I noticed that in the beginning, so I jumped in and filled that gap in a way that I learned all that, and I helped in that part the most.” (Participant 2)*


#### 3.1.3. Loan of Medical Aids

All but one NGO organize an office for lending medical aids. Medical and orthopedic aids such as beds, walkers, crutches, anti-decubitus mattresses, infusion stands, etc., can be borrowed free of charge. In this way, home care support is provided, which is especially important for those of poorer socioeconomic status.


*“Our lending office has grown over the years as we have been operating for many years, since 2010. In 2011, it has grown into a large lending office with aids that we lend free to palliative patients and that is the biggest part of what we do. We work with patients and families a lot through the office, which currently has 1000 aids. We have 70 beds, 50 wheelchairs—it is an awful lot, and so all our palliative patients in the county get the aids they need.” (Participant 7)*


#### 3.1.4. Help with Instrumental Activities of Daily Living 

The activities of some volunteer organizations are aimed at helping patients with some of the instrumental activities of daily living, such as purchasing food and medicine, going to medical examinations, or religious activities.


*“We visit our patients; take them to doctor’s appointment, to the pharmacy, to grocery shopping, up to playing board games.” (Participant 3)*



*“We visit war veterans every day, as NGO we have agreements with shopping centers through which on daily basis, we deliver food products to seriously ill veterans and those of lower socioeconomic status, in their homes.” (Participant 6)*


However, the boundary in terms of the type and scope of activities carried out by volunteers is not entirely clear. 


*“We are here to help the family and the patient. But someone can ask for, let’s say, chopping wood—we didn’t come here to chop wood, we came not to be physically used, but to give our contribution to that family and the patient.” (Participant 3)*


#### 3.1.5. Family Support, including Grieving Support

According to the participants, caring for the patient’s family is almost inseparable from caring for the patient. In children’s palliative care, the support is entirely parent-centered. Family support, in addition to the psychosocial support described earlier, includes respite care. When matching a volunteer and a patient, it is seen that the volunteer is accepted by both the patient and the family, and that the volunteer fits into the family dynamics. Such an approach is a feature of the individualized approach to each patient and the perceived importance of the family wellbeing.


*“I think we are here a lot for respite, to stay with the patient so that the family can take a break for two hours, to make it a little easier for them, to have them recharge their batteries.” (Participant 3)*



*“First it [who is the best volunteer] means who is free, second that he is somehow closer to the patient’s home, and available at a time when families need it.” (Participant 4)*


The relationship with the patient’s family continues after the patient’s death through providing support in mourning. 


*“We provide grieving support; we have three psychologists who give guidance and advice.” (Participant 5)*



*“Two support groups for grieving are currently active.” (Participant 1)*


#### 3.1.6. Raising Awareness of Palliative Care

Volunteers do not see raising awareness and understanding of palliative care as a side effect of their actions, but they approach and conduct it proactively.


*“So, we have been raising public awareness for 10 years.” (Participant 7)*



*“We have worked hard to explain to people that in fact palliative care, that there may be a need for palliative care in the treatment phase that is not strictly before death. And that people may have a need for palliative care and then go back to active treatment and so on.” (Participant 1)*


Although they confirm the progress made in this regard, they emphasize the need for further education on what palliative care is and for whom it is intended. As a target audience for education, they list health professionals, primarily doctors (working in family medicine and in the hospital), and patients’ families. Participants associate their (mis)understanding of palliative care with the patient’s late entry into palliative care and the consequent difficulties in meeting their needs. 


*“It seems to me that, no, it doesn’t seem to me, but we can certainly see progress, as time passes. Family doctors are better educated and have adopted a palliative approach, so I say, progress can be seen. However, we have difficulties and I agree with what everyone has said that people call too late. When the families and the patient, when they are already very exhausted, when they really can’t do anything anymore, when they have already exhausted all their strength, then they contact us, and in fact it’s a pity, right, because then there is not much to help them, there is little time left and we will hardly do much there. But if you start early enough, then you can really do a lot.” (Participant 4)*



*“I still think that, how can I say this—that the patient and family suffer less, even though I say they contact us late. Although, I don’t know, except that we must educate doctors, I think we should maybe educate families more, not to be ashamed to join the palliative team, not to be ashamed to ask for help...” (Participant 3)*


#### 3.1.7. Volunteer Education

Training of volunteers to perform activities in palliative care is carried out by the NGOs themselves. It includes peer-to-peer education, but also lectures by professionals working in palliative care.


*“We organize education for our members, and volunteers from other veterans’ associations who express a desire to provide palliative care.” (Participant 6)*



*“We organize courses for non-health volunteers twice a year, and there are 20 to 30 participants per course.” (Participant 5)*


### 3.2. The Position of Volunteers in the Palliative Care System

The position of volunteers in the palliative care system is determined by the organization of the system in Croatia. In particular, volunteers cover the gap in response to the needs of palliative patients between the healthcare and social care system. By their actions they point to the places and content in the care process that needs to be integrated. Volunteers have been involved in the development of the palliative care system since its inception, and the way they participate in the care process is still informal. An overview of the codes describing their position in more detail is provided in Table 2.

#### 3.2.1. Linking Healthcare and Social Care

In Croatia, health and social services are provided in parallel and fall under two different ministries. Integration at the level of resources and processes is scarce. Palliative care is regulated in the healthcare system, and volunteers are involved in its implementation. The activities they carry out are dominantly from the domain of social care. In social care regulations, persons with permanently impaired health, including palliative patients, are in principle entitled to housing, home help, and psychosocial support. However, the pace of deterioration of a palliative patient from the moment of their involvement in palliative care, and the speed of processing requests and providing social services, are in discrepancy. Apart from timeliness, it is also a question of the volume of needs covered. Therefore, sometimes a person has rights, but they are realized too slowly. In other cases, the response is quick but insufficient to cover the need. Palliative care volunteers work in this gap between the healthcare and social care systems. They compensate for what cannot be achieved on defined bases, accelerate the exercise of rights, and bridge the cracks in the system.


*“So palliative care is not just a health problem, and it is not, it does not belong to the health care system itself. Palliative care also includes the social policy or social care system. These are two systems, the two ministries that each have their own way and laws according to which they work, according to which they operate. (…) At the moment, we as NGOs are, let’s say, a kind of bridge that connects it as much as it can.” (Participant 4)*


#### 3.2.2. Networking with Health and Social Workers

The entry point to volunteer care are health and social workers who refer patients to volunteers. They are the ones who recognize the need that volunteers might respond to. They refer them by word of mouth, as there are no formal means of communication between health/social care and volunteers in place. This informal network has been established through the proactive engagement of the NGOs and their constant presence in the communities and in places where people are dying, such as oncological departments in hospitals. 


*“When we started, we did it mostly for friends, that is, for people who knew that there was an NGO, then later we advertised, and now we are very well networked with the health center, other NGOs that do similar work, with the hospital.” (Participant 1)*



*“…these are informal connections. We work with mobile [palliative] teams because they know we can help palliative patients, so they call us and work with us. We are of course very happy when we work together with them, or when we can call them when we first visit patient and see that there are health problems. So, there is no formal agreement, so to speak, some formal procedures, how we cooperate… I don’t think it would fundamentally change anything, but it might still give some weight to it.” (Participant 4)*


#### 3.2.3. Continuous Participation in the Development of Palliative Care—Filling the Gaps and Upgrading the Care Process

Palliative care in Croatia has been systematically developed since 2014, when a strategic plan on the development of palliative care was adopted. Four NGOs represented in the focus group were established and were active in palliative care even before the palliative care reform. The remaining three NGOs were established in parallel with the beginning of the development of the palliative care system in the respective county. Due to this continuity, participants have an insight into the changes that have taken place over the years in the healthcare and social care systems related to palliative care, and in the volunteer organizations themselves. From the discussion in the focus group, it was read that the activities carried out by volunteer organizations and volunteers are essentially those that: (a) are missing or insufficient; (b) are being realized too slowly; (c) have complicated procedures for their realization. With the development of the system over time, the type and scope of volunteers’ activities has changed. Volunteers perceive their activities at two ends of the spectrum: they see their work as filling the gaps in the care process, and, on the other hand, they see it as an upgrade to the existing care, in terms of a higher quality or an expanded service.


*“At the time when we started, palliative care in this county was in its infancy. At that time, it was not very clear to us how the NGO should work, what we should do, and we were simply focused on sensitizing the public, and to actually talk about it and to start things off. We wanted to draw the attention of people in the city and county administration and professionals to think together what we could do in this regard.” (Participant 1)*



*“Sometimes we fill the gaps more, sometimes it is more about complementing. Through the 10 years course, you are first one, and then you are the other. I can say that our NGO and volunteers really see themselves as serious partners because we participate in all steps of palliative care development in the county. (…) We are a complement to professional services, but we also fill in the gaps such as psychological support, such as supervision, things that should be, as others have said, continuous. We provide it when we have money for it, when we have a person or other resource and then it is really filling the gaps. But we can generally say that our community sees us as a complement to the quality and system of palliative care.” (Participant 7)*


### 3.3. Characteristics of Volunteering

Characteristics of volunteers’ action (Table 3) derive from their position in the system, but also from their motivation to participate in palliative care and to improve the patients’ experience.

#### 3.3.1. Individualized Approach

Volunteers are focused on the need expressed by patients and their families and create their response according to that need. They approach everyone individually, see a person as a whole, and do not question the reasons behind the need, nor have strict including or excluding criteria for donating their time and presence.


*“During the first visit we collect information, we take anamnesis, data about the patient and then we coordinate further. We assign a volunteer we think is best for that family, for that patient.” (Participant 3)*



*“Our biggest problem is that we meet war veterans who have mental health problems. And the only way to understand them is for the veteran to be with the veteran and then that person can, uh, loosen up and say some of their problems, which we later pass on to palliative care teams, health centers or GPs.” (Participant 6)*


#### 3.3.2. Flexibility in Response

In addition to the individualized approach, in meeting the patients’ needs, they easily overcome administrative barriers arising from the system structure and management. Furthermore, the realization of care provision is not bureaucratized. Their response is formed according to the expressed need. The flexibility is manifested in the setting, volume, or time response of care provision. 


*“Because we cover, except our county, two more neighboring counties. Because one of them, I must say, they do not even have a mobile palliative team. So, a lot of people are looking for us, a lot of family doctors as well. Because we are present and people know about us, we are available, we cover as much as we can.” (Participant 3)*



*“Palliative patient does not have time to wait for their [the Ministry’s] answer because if we can’t do it now, at this moment, if we wait for an answer for fifteen days, it’s worth nothing. We must act now… Therefore, we have to act immediately on our own. So, we are looking for any contacts with all other institutions to resolve it now and not to wait for answers.” (Participant 6)*


#### 3.3.3. Presence in the Community

Volunteer organizations are deeply involved in the community in which they operate. They adapt to its dynamics, and it makes them recognizable and approachable. However, they are mostly located in urban regions, and the access to their support is poor in rural or isolated areas of their counties.


*“So we are at all the markets, fairs and so on, so that they [the community] know about us.” (Participant 3)*



*“We strive to constantly work on visibility, and then often families and patients, let’s say, while they google where to go, what to do, they come across us, and they contact us in different ways.” (Participant 4)*



*“I would agree with Participant 5 that we actually miss people, we really miss people because we gravitate to both islands and hinterlands, and it actually happens to us that we cannot cover and provide services to people who need it because we cannot technically and often physically reach all these places and provide as much as there are needs.” (Participant 1)*


#### 3.3.4. Project Funded 

Volunteer work is organized through non-profit NGOs. Although the work of the volunteers themselves does not include any financial or other compensation, NGOs have a number of employees on a salary, who lead the organization and perform some of the activities. In particular, they recruit, educate, coordinate, and supervise volunteers. NGOs are financed by donations and, to a lesser extent, by membership fees. Donations come from individuals and legal entities. Among the public legal entities are the Ministries of Health, Social Care, and Veterans; Local Self-Government Bodies such as County and City Offices for Health, Social Care, and Veterans; and the National Foundation for Civil Society Development. Some donations come through the investment programs from European Union funds, e.g., the European Social Fund. Donations from these institutions are received based on applications for a public tender for financial support in the implementation of certain types of activities, or activities aimed at specific population groups. In the focus group, this funding model was described as uncertain, of questionable sustainability, and inappropriate given the perceived importance of the role that NGOs and their volunteers play in palliative care.


*“We had psychological support and supervision projects for professionals, but that part was realized through projects, and COVID was a bit of a hindrance. (…) We pay the supervisor for professionals, but we are only able to provide support if we raise enough money.” (Participant 7)*



*“So, we are supplementing the system, but we are financed outside the system. We are trying, with our skills and knowledge, to prepare how to read tenders and pass tenders, that is how we are financed. So, in that sense it is not, it is not a system, it is not, it is not a systematic solution if we have to apply for projects every year and pray to God that we pass.” (Participant 4)*


## 4. Discussion

The main finding of our study shows the dichotomy of volunteers’ role and position in the Croatian palliative care system. It is manifested in the fact that in relation to the palliative patient, they are regulated through the healthcare system, while primarily providing social care. Next, although a system that is highly regulated relies on their participation in care, they are involved in the care process informally and are flexible in their approach and response. Lastly, their position is to fill in the gaps where the system is not adequately or sufficiently developed, and, on the other hand, they are the drivers of innovation in care, developing new forms and types of care. In this section we discuss these seemingly contradictory perspectives further.

### 4.1. Being Formal or Informal Care Providers

Volunteers are regulated within the healthcare system in terms of education in palliative care and settings in which they may perform activities, and are listed as associates in multidisciplinary teams. Although their formal involvement in multidisciplinary palliative care teams is envisaged, primarily through the establishment of communication channels and the exchange of patient data and information, this has not been achieved so far. They communicate informally with healthcare professionals and are included in the care process by word of mouth. Once volunteers enter the palliative care system, their activities are not regulated by ordinances, financially or otherwise, but are tailored to the specific needs of the individual patient and family. Looking from the process perspective, this informality allows volunteers to fit into the existing care process when and where they are required and to the needed extent. On the other hand, formalizing the role of volunteers would include a clear protocol of who is a volunteer in care, when, why, and under what conditions volunteers in care are included, and what resources they use. Such protocols are important for the standardization and sustainability of care, in relation to patient quality and safety [38]. This duality, with its advantages and disadvantages, is described by McLennan, Whittaker, and Handmer [39]. Naylor et al. state volunteers’ involvement in multiprofessional teams as being a risk to their flexibility and independence [15]. The potential loss of flexibility and need-oriented services is also discussed by Bode and Brandsen, in the context of partnerships between the state and NGOs [40]. Furthermore, the question addressed in research on volunteers related to the formalization of their role is how rules and oversight might negatively affect the intrinsic motivation of the volunteers themselves [39]. Additionally, as volunteers generally donate their time and are there to be present, there is a question of the line between volunteering and unpaid work, in the case of further formalization of volunteers’ roles in long-term care [41,42]. The second question is whether volunteers should complement or substitute the work of professionals [43,44]. In the perspectives described, they seem to do both.

### 4.2. Between Healthcare and Social Care

Palliative care legislation does not explicitly state the content of patient-oriented activities carried out by volunteers. Results from the focus group show that the content of care provided by volunteers is mainly from the domain of the social needs of patients. In healthcare systems, professionals provide care, whereby the content and scope are defined in accordance with the health indication and paid for on that basis [45]. The social care system has its own criteria for the engagement of professionals and the exercise of rights in this area. Among the leading criteria are the census on money and income and disability status [28]. The combination of both types of services, health and social, maintains a patient’s quality of life to the end [8]. 

Due to the pronounced health component towards the end of life, the patient mostly leans on the health system. Unlike aging, which is a slow process, for a palliative patient a rapid deterioration (physical, mental, functional) is characteristic in the last year of life [46]. Therefore, in the case of the palliative patient, the social care system cannot respond to these needs as quickly as they occur, especially if the identification of such patients is delayed and they are included in palliative care late (several months or weeks before death). Thus, a gap of unmet (in general, untimely) needs opens between the two systems, in which there are volunteers in palliative care. Faulkner and Davis explored and described the role of volunteers precisely through the provision of social support in healthcare [47]. 

In Croatia, where palliative care is integrated into the healthcare system, our results show that volunteers act as an extended arm of the social care system in the healthcare system to ensure continuity of care and support the integration of care for patients with complex needs. None of the documents indicate that this is a strategically envisaged role of volunteers but emphasize the need for the integration of the health and social systems. However, there is no clear idea given of what needs to be integrated. Therefore, we can say that the nature of volunteers’ activities accentuates the problem of this unnatural division of the health and social aspects of a patient (person) into two administrative systems, i.e., ministries with separated or parallel processes and inconsistent terminology, because of which they find it difficult to communicate with each other. Moreover, by observing their activities, we begin to understand what needs to be integrated. Taels et al. tackle social work in palliative care by addressing the prerequisites for their meaningful involvement [48].

### 4.3. Filling the Gaps and Piloting Innovation

Volunteering in palliative care in the Republic of Croatia began before the systematic development of palliative care within the healthcare system [11]. With the development of the system, the dual role of filling the gaps in care and piloting innovations they describe, changed the ratio more towards the latter. Process-wise, when filling the gaps, volunteers organize their resources as requested and fit them into existing care processes where there is a missing or insufficient service, a service being realized too slowly, or a service that has complicated procedure for its realization. When upgrading the existing care, in terms of a higher quality or an expanded service, a completely new process is developed, which may or may not anticipate other stakeholders. By recognizing the need and shaping the response, volunteers contribute to the development of social innovation [49,50,51].

The literature shows a difference between the roles of volunteers in low- and high-income countries. Where the palliative care system (and healthcare in general) is less developed, volunteers even carry out some nursing procedures [52,53,54]. In high-income countries, with more available and accessible health and social care systems, volunteers are more likely to keep patients company and just be present [20]. It follows that the role of volunteers and their activities might indicate weaknesses and shortcomings in the formal care process and point to where it is necessary to improve the content or format of care provision.

Some countries already have strategies for the engagement of volunteers, meaning that they are in a way formalizing and regulating their role and position [15,39]. Another approach is that the state uses its mechanisms to take over what volunteers do, and integrates it into the existing paid services or develops new ones. After all, this happened with palliative care. The question that arises is what the minimum standard of formally provided services should be, and whether (financial) sustainability and the level of quality can be ensured as the standard increases over time. 

Lastly, palliative care was initially defined to respect the whole person, including the physical, social, psychological, and spiritual aspects of the ill person [7]. However, as it developed predominantly in healthcare, the medical part, aimed at alleviating pain and other symptoms, has been most pronounced. In recent years, a shift has occurred, in which palliative care is increasingly viewed through the prism of a public health approach and a social model of health, as opposed to a disease-oriented model. This means working at the community level, with the aim of improving peoples’ experience of dying, death, and bereavement [55]. Such communities are called compassionate communities [56,57]. Given the trust volunteers have gained, their achievements in raising awareness of palliative care in the community, and social support they already provide, they could be key partners in building compassionate communities and thus help provide appropriate palliative care for all.

### 4.4. Limitations

The study comprised one focus group with seven participants. Looking at the number of NGOs active in palliative care, we achieved a representation of more than 50% [34]. However, it remains unknown as to what extent the findings are applicable to the whole population of volunteers. The focus group was conducted via Zoom. The advantages and disadvantages of Zoom as a platform for data collection in qualitative analysis are considered because it has only recently begun to be used as such [58,59]. Of the advantages mentioned, we were able to include participants from different parts of the country, while they might not travel a few hundred kilometers to participate in a face-to-face focus group. Among the challenges, technical difficulties and the familiarity of participants with technology are the biggest mentioned. In this sense we had no difficulties because the participants were already familiar with the use of Zoom.

## 5. Conclusions

Volunteers seem to be one of the key stakeholders in providing palliative care and are recognized by professionals and the community. The informality and independence of their position as a non-governmental sector allow them flexibility and an individualized approach in response to the specific needs of individuals and their families. Nevertheless, with it come some uncertainties, especially in terms of sustainability.

Volunteers bridge the gaps in the formal provision of care within the healthcare system, especially where links between healthcare and social care have not been established or are broken. Therefore, they might indicate what needs to be integrated between the two systems and where.

Their role changes with the development and improvement of a formal palliative care system, as their focus quickly shifts to the emerging unmet needs, around which they develop new activities.

## Figures and Tables

**Table 1 ijerph-19-08203-t001:** Codes within a theme. The role of volunteers in palliative care.

Codes
Psychosocial support
Assistance in exercising rights
Organization of lending aids
Help with instrumental activities of daily living
Family support, including grieving support
Raising awareness of palliative care
Volunteer education

**Table 2 ijerph-19-08203-t002:** Codes within a theme. The position of volunteers in the palliative care system.

Codes
Linking healthcare and social care
Networking with health and social workers
Continuous participation in the development of palliative care
Filling the gaps in the care process
Upgrading the care process

**Table 3 ijerph-19-08203-t003:** Codes within a theme. Characteristics of volunteering.

Codes
Individualized approach
Flexibility in response
Presence in the community
Project funded

## Data Availability

Not applicable.

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
