# Peer review of "A Qualitative Study on the Position and Role of Volunteers in Integrated Care—An Example of Palliative Care in Croatia"

_ijerph, 2022, doi:10.3390/ijerph19138203_

Round 1
Reviewer 1 Report
I commend the authors for sharing their work in this important area - this is an excellent paper.
There is just one key omission that is important to address through minor revision - the EAPC white paper has been cited but there is no acknowledgement of the EAPC Charter for Volunteering in Hospice and Palliative Care, or discussion of it's potential impact (or lack of impact) in the Croatian context. This is relevant because Croatia was a country involved in developing the Charter, and there has been come critique that the Charter lacked practical application to be truly useful in its aims. This may help your introduction section, as from my review, the introduction did not provide sufficient background and include all relevant references.
I would suggest that discussion of this would be well supported by the work of Vanderstichelen et al. (2021) - in their paper 'Evaluating the EAPC Madrid Charter on volunteering in hospice and palliative care: Reflections on impact'
https://www.tandfonline.com/doi/full/10.1080/09699260.2021.1964678
Please note that this is *not* my work - I am not one of the authors.
With this gap addressed, I would be happy to recommend it for publication in the journal. I believe it will be of interested to the readership and will be well-cited in future research.
Reviewer 2 Report
Dear authors,
thank you very much for this interesting manuscript! I find the topic interesting.
Before it can be published you should revise the paper based on my following comments:
1) Please consider to change the title - the title should cover the topic and provide some basic information about the design/methods used
2)The introduction is too long. Please make it shorter
3) The method section needs to be described in more detail. Youn have used just one Zoom focus group with 7 informants. Please write more about the qualitative method used. In my opinion your source 33 is not enough to understand what you have Donne. Please explain the analysis of the data in more detail.
4) Limitations are totally missing! You should include a limitations section. Limitations that must be included and discussed are: -only one focus group, -just 7 participants, -use of Zoom in qualitative research (that should also be discussed in a method-discussion in the discussion section.
5) Results: The results should be more structured and with clear subheadings. You use too many codes from my point of view. This could be as sign that your analysis is not totally finished yet. I recommend to look critically through your data and analysis again in order to question your findings and probably to reduce the number of codes. Probably your results will be easier to understand if they are shorter and more structured.
6) Diskussion: I Fouls appreciate a short summary of the main findings at the start of the discussion. This could make ir easier for the reader to follow and understand your thoughts. Some parts could be shorter and/or include citations to support your claims.
Good luck with a revision of your manuscript!
